# Degradation of Tetracycline Hydrochloride by Cu-Doped MIL-101(Fe) Loaded Diatomite Heterogeneous Fenton Catalyst

**DOI:** 10.3390/nano12050811

**Published:** 2022-02-28

**Authors:** Kang-Ping Cui, Yu-Ying He, Kai-Jie Xu, Yu Zhang, Chang-Bin Chen, Zheng-Jiang Xu, Xing Chen

**Affiliations:** 1School of Resource and Environmental Engineering, Hefei University of Technology, Hefei 230009, China; heyuying131@163.com (Y.-Y.H.); XUkaijie@mail.hfut.edu.cn (K.-J.X.); 2Key Lab of Aerospace Structural Parts Forming Technology and Equipment of Anhui Province, Institute of Industry and Equipment Technology, Hefei University of Technology, Hefei 230009, China; zhangyu70714@163.com; 3Anqing Shuguang Chemical Co., Ltd., Anqing 246003, China; chenchangbin19@163.com (C.-B.C.); xuzhengjiang19@163.com (Z.-J.X.)

**Keywords:** Fenton-like catalyst, bimetallic MOF, diatomite, synergistic effect, antibiotics

## Abstract

In this work, the combination of high surface area diatomite with Fe and Cu bimetallic MOF material catalysts (Fe_0.25_Cu_0.75_(BDC)@DE) were synthesized by traditional solvothermal method, and exhibited efficient degradation performance to tetracycline hydrochloride (TC). The degradation results showed: Within 120 min, about 93% of TC was degraded under the optimal conditions. From the physical–chemical characterization, it can be seen that Fe and Cu play crucial roles in the reduction of Fe^3+^ because of their synergistic effect. The synergistic effect can not only increase the generation of hydroxyl radicals (•OH), but also improve the degradation efficiency of TC. The Lewis acid property of Cu achieved the pH range of reaction system has been expanded, and it made the material degrade well under both neutral and acidic conditions. Loading into diatomite can reduce agglomeration and metal ion leaching, thus the novel catalysts exhibited low metal ion leaching. This catalyst has good structural stability, and less loss of performance after five reaction cycles, and the degradation efficiency of the material still reached 81.8%. High performance liquid chromatography–mass spectrometry was used to analyze the degradation intermediates of TC, it provided a deep insight of the mechanism and degradation pathway of TC by bimetallic MOFs. This allows us to gain a deeper understanding of the catalytic mechanism and degradation pathway of TC degradation by bimetallic MOFS catalysts. This work has not only achieved important progress in developing high-performance catalysts for TC degradation, but has also provided useful information for the development of MOF-based catalysts for rapid environmental remediation.

## 1. Introduction

In recent years, the problem of wastewater caused by antibiotics has been widely concerning. With the development of modern medicine, antibiotic pollution is becoming one of the water pollution problems faced by all humankind [1]. Because many antibiotics are not biodegradable and have strong chemical stability in water environments, the antibiotics in municipal wastewater especially cannot be completely removed, which endangers the safety of the ecosystem and causes great harm to human health [2,3]. The annual discharge of antibiotics in China exceeds 50,000 tons, and the ecological environment of most water bodies is seriously threatened. As a typical antibiotic, tetracycline hydrochloride is widely used in human and veterinary medicine [4,5]. Among different antibiotics, tetracycline hydrochloride (TC) occupies second place in production and usage, and has high antibacterial activity against a variety of pathogenic bacteria [6]. Therefore, it is important to explore new techniques for the effective degradation of tetracycline, especially Fenton oxidation, for TC removal from wastewater.

At present, Fenton oxidation is the most-used method for wastewater with high antibiotic content. However, the traditional Fenton oxidation based on Ferrous ions (Fe^2+^) suffers from some serious drawbacks, including the production of large amounts of Fe sludge, a narrow pH range and exhibiting low reusability [7,8]. To address the shortcomings of homogeneous Fenton, a heterogeneous Fenton process using solids as catalysts was developed, such as γ-FeOOH/GPCA [9], MnFe_2_O_4_/bio-char [10], and Fe_3_O_4_/SiO_2_ [11].

Metal–organic frameworks (MOFs) are crystalline porous materials formed by the assembly of organic ligands and transition metal ions (or clusters). MOFs have developed rapidly in the past two decades and have impacted many fields, including separation [12], energy storage [13], sensing [14], and catalysis [15]. MOF materials have been widely used in catalysis over the past few decades due to their unique structure. MOFs have many useful properties, such as tunable porosity, interconnected pores, and large specific surface area. These properties facilitate the diffusion of pollutants and products during catalytic reactions. Fe-based MOFs (Fe-MOFs) have received particular attention over the past few decades [16,17]. Compared with other metal catalysts, Fe is non-toxic and the second-most productive metal. Although some other transition metals (such as Co [18]) have been gradually applied to heterogeneous Fenton catalysts, the high toxicity and high application cost of these metals make the Fe-based heterogeneous catalysts stand out from other catalysts.

In recent years, it has also been noticed that when a second metal ion invades the framework nodes, the catalytic performance of MOFs can be improved [19]. Bimetallic catalysts have excellent catalytic performance because of the increased specific surface area of the material or the synergistic effect between metals, which also provides additional reaction sites for H_2_O_2_ activation. Due to the good compatibility of MOFs with various metal centers (such as Fe, Cu, Co, and Zn), some MOFs can have both metal centers and bind to organic ligands [20,21], thus the catalytic activity of heterogeneous reaction was improved. For example, Qiao Sun et al. first described partial substitution of Fe in Fe-(BDC) metal–organic frameworks with Mn, Co, and Ni. The results show that the addition of Mn can significantly improve the degradation efficiency [22]. Bimetallic Fenton catalysts prepared by using MOF materials as precursors or templates have better catalytic performance than traditional preparation methods. Some researchers have synthesized CoMn_2_O_4_ microporous plates using MOFs as precursors. Compared with CoMn_2_O_4_ synthesized by the traditional solvothermal method, this catalyst has a higher specific surface area and abundant •OH_surf_. It also has higher catalytic activity [23]. The results show that the use of bimetallic MOFs as efficient Fenton-like catalysts is a promising approach.

Because the redox potential of Cu^2+^/Cu^+^ (E_0_ = 0.17 V) is lower than Fe^3+^/Fe^2+^ (E_0_ = 0.77 V), Cu^+^ has a stronger reducibility than Fe^2+^. Meanwhile, the interfacial electron transfer can be accelerated through the interaction between the two metal redox pairs of Fe and Cu, thereby improving the catalytic performance of H_2_O_2_ activation [24]. In particular, the Cu has Lewis acid properties, which can interact with Fe species to form a local acidic microenvironment, and expand the pH range of the system [25]. Modification of pristine MIL-101(Fe) with copper as the second metal component and optimized the proportions of Fe and Cu metal elements. 

In addition, the activity and stability of the catalyst decreases due to the inevitable metal leaching and agglomeration of nanoparticles [26]. In addition to doping non-metallic elements [27,28], fixing the MOFs in the core–shell [29,30] or loading it on the macromolecular carrier [16,31] can also reduce the leaching of metal ions and make the metal ions disperse and not agglomerate easily. Diatomite (DE) is a naturally-occurring porous mineral with a complex three-dimensional (3D) structure [32]. Acid treatment may increase the surface area and pore size by eliminating impurities responsible for pore blockage [33]. DE has good adsorption performance and is a highly promising potential carrier material candidate for heterogeneous Fenton catalysts. Using DE as the carrier of the metal catalyst can decrease their aggregation effect and thus improve the catalytic activity [34]. Simultaneously, the stability of the catalyst can also be improved by reducing the leaching of metal ions [16].

In this study, Fe_0.25_Cu_0.75_(BDC)@DE was fabricated and used as a Fenton-like catalyst to degrade TC in a solution. Above all, the ratio of Fe and Cu was optimized, and the catalyst reached a high level of degradation. Then the optimal ratio of the catalyst was loaded onto diatomite to reduce agglomeration and metal ion leaching. Finally, through detailed characterization of the composite catalyst, including the morphology, composition, and other structural properties of the catalyst, the stability and cyclability of the catalyst and the possible catalytic mechanism and TC degradation pathway were elucidated.

## 2. Experimental

### 2.1. Materials and Chemicals

The chemicals used in this work were iron (Ⅲ) chloride hexahydrate (FeCl_3_·6H_2_O), copper nitrate trihydrate (Cu(NO_3_)_2_·3H_2_O), sulfuric acid (H_2_SO_4_), and potassium iodide (KI), all acquired from Sinopharm Chemical Reagent Co., Ltd., Shanghai, China. Terephthalic acid (H_2_BDC), anhydrous ethanol (C_2_H_5_OH), tetracycline hydrochloride (TC·HCl), p-Benzoquinone (BQ), and tertiary butanol (TBA) were supplied from Shanghai Maclin Chemistry Co., Ltd., Shanghai, China. Hydrogen peroxide (H_2_O_2_), *N*, *N*-dimethylformamide (DMF), and sodium hydroxide (NaOH), were purchased from Shanghai Aladdin Chemistry Co., Ltd., Shanghai, China. Diatomite (DE) was purchased from International Building Materials Environmental Technology Co., Ltd., Shanghai, China.

### 2.2. Preparation of Catalysts

#### 2.2.1. Synthesis of Fe_x_Cu_y_(BDC)

Fe and Cu bimetallic MOF composites were produced via a facile solvothermal method. In a traditional synthesis, the required number of FeCl_3_·6H_2_O and Cu(NO_3_)_2_·3H_2_O was added to 30 mL of DMF solution, and then 5 mmol H_2_BDC (0.830 g) was dissolved in 30 mL DMF of solution under magnetic stirring. After mixing the two solutions and continuing the magnetic stirring for half an hour at room temperature, the solutions were poured into a 100 mL Teflon-lined stainless-steel autoclave (Lichen, Shanghai, China), and the temperature was maintained at 110 °C for 36 h under autogenous pressure for a solvothermal reaction. After natural cooling, the resulting precipitates were collected and washed with DMF, ethanol, and deionized water to remove residual impurities, and dried at 80 °C for 12 h. The total molar amount of FeCl_3_•6H_2_O and Cu(NO_3_)_2_•3H_2_O used to synthesize the samples was fixed at 10 mmol. The resultant samples were donated as Fe_0.75_Cu_0.25_(BDC), Fe_0.5_Cu_0.5_(BDC), and Fe_0.25_Cu_0.75_(BDC), separately. For comparison with bimetallic MOFs, two monometallic samples containing only FeCl_3_•6H_2_O and Cu(NO_3_)_2_•3H_2_O were prepared: Fe(BDC) and Cu(BDC). Table 1 records the parameters of all sample synthesis.

#### 2.2.2. Synthesis of Fe-Cu Bimetallic MOF@DE

Fe-Cu bimetallic MOF@DE were produced using the same operations procedures, the solution was mixed, and 1 g of DE was added and magnetic stirring continued for 1 h. It was then poured into a 100 mL Teflon-lined stainless-steel autoclave, and the temperature was maintained at 110 °C for 36 h under autogenous pressure for a solvothermal reaction.

### 2.3. Characterization of Catalysts

The specific surface morphology of all the prepared samples were obtained by SU-8020 scanning electron microscopy (SEM; Hitachi, Tokyo, Japan) and JEM-2100F transmission electron microscopy (TEM; JEOL, Tokyo, Japan). The crystalline structures of the prepared samples were recognized by powder X-ray diffraction (XRD) measurements (PANalytical, Almelo, The Netherlands). The XRD patterns were documented in the 2θ range of 5–90° with a scan rate of 0.06° min^−1^ working at 40 kV and 40 mA. The nitrogen adsorption–desorption isotherm at 77 K was used to analyze the specific surface area and pore size distribution of samples by the Brunauer–Emmett–Teller (BET) analysis (Autosorb-IQ3, Quantachrome, Boynton Beach, FL, USA). Fourier transform infrared (FTIR) spectra was recorded by KI suppression disk technique in the 4000–400 cm^−1^ wave number range with a resolution of 2 cm^−1^ (KQ-300DE, Thermo, Waltham, MA, USA). The chemical valence states of the elements contained in the sample were detected by X-ray photoelectron spectroscopy (XPS, Shimadzu, Kyoto, Japan).

### 2.4. Catalytic Degradation Experiments

The reaction was added to a 100 mL sealed serum bottle with 50 mL of 20 mg/L TC solution, 0.5 M NaOH, and H_2_SO_4_, which was used to adjust the initial pH value. the samples was then placed in a constant temperature shaker table (200 rpm, 25 °C) for adsorption to reach equilibrium after 30 min. H_2_O_2_ was then added for a Fenton-like degradation reaction. During the reaction, 1 mL of the solution was collected at given time intervals and filtered to remove solid impurities. After quenching with excess tert-butanol, the absorbance of the pollutant was tested at 375 nm using an ultraviolet spectrophotometer. For the recycling measurement, the catalyst was collected by centrifugation, the material washed until neutral, and then dried.

The stock solution of TC was prepared with a concentration of 1 g/L, diluted to different concentrations, and its standard curve was measured. The mother liquor was diluted to the corresponding concentration and we checked whether its concentration was accurate every time the degradation experiment was performed to ensure that the TC was not photolyzed.

### 2.5. Analytical Methods

TC contents were measured by ultraviolet spectrophotometer (UV–Vis; Hitachi, Tokyo, Japan). We detected the dynamic change of absorbance in the wavelength range during the reaction was 200–450 nm. The degradation intermediates of TC were assayed by high performance liquid chromatography–mass spectrometry (HPLC-MS; Water, Williamsburg, VA, USA). The mobile phase was a mixture of acetonitrile and oxalic acid (20:80, *v/v*) with a flow rate of 1.0 mL/min. Total organic carbon (TOC) was analyzed by using a TOC/TN analyzer (Multi N/C 2100, Analytik Jena AG Corporation, Jena, Germany). The amount of leached Fe and Cu ions in the solution after the reaction were detected by atomic absorption spectroscopy (AAS; AA140, VARIAN, Crawley, UK). •OH radicals were detected with 50 mM DMPO as the scavenger using electron paramagnetic resonance spectroscopy (ESR; Bruker, Bremen, Germany).

## 3. Results and Discussion

### 3.1. Characterization of Catalysts

The crystal information and chemical composition of the samples prepared with different ratios of Fe and Cu were analyzed by XRD. Figure 1 showed the XRD spectra of Fe and Cu ratios of 1:0, 3:1, 1:1, 1:3, and 0:1, respectively. For the diffraction peak of Cu-free doped catalyst, the value of the obvious peak is consistent with the diffraction peak of MIL-101 (Fe) previously reported [35]. When Cu is added into the catalyst, the ratio of Fe to Cu becomes 3:1. The diffraction peak of the sample has no significant difference, indicating that adding small amounts of Cu to the catalyst did not affect the type of material produced, but the existence of Cu complexes is undeniable. Then the doping amount of Cu continues to increase and the ratio of Fe to Cu becomes 1:1, 1:3, and 0:1, and the diffraction peaks among the three have no obvious difference. The main diffraction peaks of Cu(BDC) were basically consistent with those reported before [36]. After being doped with Cu, the XRD spectra of the four bimetallic samples are similar to MIL-101(Fe), indicating that Cu was successfully incorporated into the MIL-101(Fe). After doping the catalyst with the optimal ratio and diatomite, it can be seen from the diffraction peaks in the figure that the crystal structure of the sample has changed after adding Cu, so the XRD characterization results are also different. Different catalyst materials can be obtained by adjusting different ratio of Fe and Cu, and the degree to the left is less than 1°.

The morphological structures and microstructures of the prepared samples were observed by SEM and TEM. Through SEM analysis, the morphological structures of these prepared samples were discerned. The FESEM image of Fe(BDC) exhibits an octahedral crystallite structure, while Cu(BDC) shows the presence of a cubic layered structure [37]. As shown in Figure 2a, there is disintegration of the original octahedral morphology into small irregular particles due to the introduction of Cu into the MIL-101(Fe) framework. It showed that Fe_0.25_Cu_0.75_(BDC) grew in a two-dimensional direction, forming irregular sheet-like structures. DE shows the perforated regular disc morphology (Figure 2c,d). It can be seen in (Figure 2b) where the Fe_0.25_Cu_0.75_(BDC) decorated on the DE surfaces results to form Fe_0.25_Cu_0.75_(BDC)@DE composite materials. From the TEM images (Figure 2e,f), the Fe_0.25_Cu_0.75_(BDC) were uniformly distributed. Simultaneously, the TEM showed Fe_0.25_Cu_0.75_(BDC)@DE: these components of Cu and Fe were equally formed on the DE surface. Furthermore, the SEM elemental mapping of Fe_0.25_Cu_0.75_(BDC)@DE revealed that these components of Fe, C, and O were uniformly formed on the surface (Appendix A).

Figure 3a displayed FT-IR spectrum of these synthesized catalysts. The O-H stretching vibration of water molecules adsorbed on the catalyst surface forms a broad band centered at 3430 cm^−1^. Characteristic peaks at 1570 and 1393 cm^−1^ are due to carboxylate vibration. The characteristic absorption bands at 1090 cm^−1^ and 796 cm^−1^ can be attributed to the asymmetric stretching vibration and symmetric stretching vibration of the Si-O-Si bond [38], respectively. The stretching vibration of Fe-O corresponds to 542 cm^−1^, while the characteristic peak at 569 cm^−1^ belongs to the Cu-O stretching vibration mode [21]. After the synthesis of the two metals at the same time, the wide peak is offset to 560 cm^−1^, which further proves the formation of bimetallic catalyst. 

Figure 3b shows the nitrogen adsorption–desorption isotherms and pore size distributions of the composite catalysts. At moderate relative pressures (from 0.2 to 0.9), the N_2_ uptake of the catalyst samples gradually increases, indicating the presence of mesopores in the catalyst structure. The Fe(BDC) sample has higher microporosity because it exhibits significant N_2_ uptake at lower relative pressures (below 0.05) [37]. With the incorporation of Cu, the catalyst changes from micropores to mesopores, and the nanopore collapse is caused by the incorporation of Cu. The Fe_0.25_Cu_0.75_(BDC) and DE surface areas were severally 41.24 m^2^/g and 37.13 m^2^/g. After doping, the specific surface areas of the composites were reduced to 30.81 m^2^/g, which further proved that Fe_0.25_Cu_0.75_(BDC) was supported with DE (Table 2). The DFT pore size distribution curve also showed that the pore size of the composite catalyst was distributed in both mesoporous and microporous, but both were smaller than that of Fe_0.25_Cu_0.75_(BDC) and DE.

The chemical states of Cu and Fe on the composite catalyst surface can be studied by XPS survey. As observed from Figure 4a, four peaks at 711.0, 712.9, 724.6, and 726.5 eV in XPS spectra, assigned to Fe^2+^ 2p_3/2_, Fe^3+^ 2p_3/2_, Fe^2+^ 2p_1/2_, and Fe^3+^ 2p_1/2_, indicating that the composite catalyst is composed of Fe^2+^ and Fe^3+^ species [25]. In the Cu2p diagram (Figure 4b), four peaks are fitted at 933.2, 934.8 eV, 953.0, and 954.6 eV, corresponding to Cu^+^ 2p_3/2_, Cu^2+^ 2p_3/2_, Cu^+^ 2p_1/2_, and Cu^2+^ 2p_1/2_, which are the characteristic peaks of Cu^2+^ and Cu^+^ species [39]. The kinetic energy (KE) peaks in Figure 4c are 915.6 eV and 917.6 eV, corresponding to Cu^+^ and Cu^2+^, respectively. The results further illustrate that the Cu species on the surface of Fe_0.25_Cu_0.75_(BDC) and Fe_0.25_Cu_0.75_(BDC)@DE samples mainly exist in the form of Cu^2+^ and Cu^+^ The results show that the polyvalent states of Fe (Fe^2+^ and Fe^3+^) and Cu (Cu^+^ and Cu^2+^) coexist in composite catalyst. The survey spectra for Fe_0.25_Cu_0.75_(BDC) and Fe_0.25_Cu_0.75_(BDC)@DE were also assessed by the characteristic elements such as trace amount of C 1 s and major elements O 1 s and Si 2p scan (Appendix A).

### 3.2. Catalytic Performance of Catalysts

According to the adjustment of different ratios of Fe and Cu, the optimal ratio of Fe and Cu is selected as shown in Table 1, and then the catalyst with the optimal ratio is loaded onto diatomite to reduce the agglomeration of materials and the leaching of metal ions. The degradation efficiencies of five different Fe/Cu ratio catalysts at an initial TC concentration of 20 mg/L were compared in the Fe_x_Cu_1−x_(BDC)/H_2_O_2_ system (Figure 5). Almost no TC removal was observed when H_2_O_2_ was added alone, revealing that the contribution of H_2_O_2_ auto-oxidation is negligible. Simultaneously, the catalyst was first added to the TC solution for 30 min to achieve equilibrium of adsorption, and then H_2_O_2_ was added for Fenton-like reaction. As displayed in Figure 5, after 30 min of adsorption equilibrium, the degradation effect of Fe_0.25_Cu_0.75_(BDC) reached more than 50% at the 5 min, which was far greater than that of the other four catalysts. After 60 min, the removal efficiency of Fe_0.25_Cu_0.75_(BDC), Fe(BDC), and Fe_0.75_Cu_0.25_(BDC) was more than 90%. The removal efficiency of Cu(BDC) and Fe_0. 5_Cu_0. 5_(BDC) to degraded TC is only 10% and 60%, respectively. In addition, As the Cu content increases, the pore size distribution of Fe_0.25_Cu_0.75_(BDC) changes from micropore to mesopore, which may be due to the collapse of nano-pore caused by Cu incorporation [37].

The enlarged pore structure will facilitate the transfer and contact of reactants to the internal reaction site, and the appropriate pore distribution provides ideal permeability. At the same time, Cu has the property of Lewis acid, which can generate a local acidic microenvironment through interaction with Fe species, thus improving the degradation efficiency of the material. In addition, the reason for the improved catalytic activity might be that the addition of Cu promotes the formation of Fe in the FexCu_1−x_(BDC) structure. The interfacial electron transfer can be accelerated by the interaction between the redox pair of Fe and Cu, thus improving the performance of the catalyst to activate H_2_O_2_. After increasing the amount of Cu, the pore structure collapse leads to a significant reduction in the specific surface area of the material, resulting in a limited number of active sites for activation of H_2_O_2_. With the increase of Cu doping, Lewis acids of Cu accounted for most of the influence, which also explained why the material degraded 50% TC within 5 min.

In order to reduce the aggregation of MOFs and the leaching problem of metal ions, the catalyst material with the best degradation effect can be loaded on the diatomite. It can be seen from Figure 6 that the degradation effect of the catalyst has reached more than 80% after the material is loaded on diatomite for 1 h of catalytic reaction. Even though the content of metal and the leaching of metal ions in Fe_0.25_Cu_0.75_(BDC)@DE are less than those in Fe_0.25_Cu_0.75_(BDC), it has no effect on the removal effect of the catalyst. This further proves that the heterogeneous Fenton reaction plays a dominant role.

To optimize the experimental conditions, we further researched the effects of pH, H_2_O_2_ concentration, and catalyst dosage on the removal effect of TC by the composite catalyst. As the catalyst loading increased from 0.1 g/L to 0.7 g/L, the degradation rate of TC increased from 74.7 to 86.9%. This may be related to the increase of active sites activated by H_2_O_2_. From Figure 6a, with the catalyst dosage increasing, the removal rate of the contaminates was augmented. However, when the catalyst dosage was increased from 0.5 g/L to 0.7 g/L, the rate of removal efficiency slowed down. This was a result of the aggregation of nanoparticles and diffusion limitation at higher concentrations of catalyst [40]. Therefore, 0.5 g/L was the optimum catalyst dosage and was chosen for further experiments. 

As shown in Figure 6b, the degradation effect was significantly enhanced when the H_2_O_2_ concentration was increased from 2 mM to 8 mM. TC removal rate increased from 70.6 to 87.2% when hydrogen peroxide dosage increased to 0.6 mM. This is probably due to the higher H_2_O_2_ dosage which could be activated by the composite catalyst and generate more •OH radicals to degrade TC. Nevertheless, in further increasing the H_2_O_2_ dosage to 10 mM, the TC removal efficiency was reduced which was probably owing to the excessive H_2_O_2_ causing the seif-scavenger effect of •OH radicals (Equations (1) and (2)) [21,41]. Thus, the optimum dosage of H_2_O_2_ was 8 mM.

It is well acknowledged that initial pH value has a significant effect on the surface charge, existing form, stability of the catalyst, etc. [17]. Therefore, the influence of the initial pH with the range from 3 to 9 on the degradation of TC were investigated and the results are shown in Figure 6c. When the pH is 3, the removal efficiency of TC is 93.0%, which is consistent with the high removal efficiency of heterogeneous Fenton under extremely acidic conditions. When the pH is 9, the removal efficiency of TC also reaches 91.3%, but it is related to the adsorption properties of DE. Under alkaline conditions, the adsorption effect of DE is better. However, the adsorption effect of alkaline conditions is not conducive to the reuse of materials. When the pH is 5 and 7, the removal efficiency of TC also reaches about 80%, indicating that the material has a good degradation effect on TC under acidic and neutral conditions, which greatly expands the removal range of the pH. This may be due to the Lewis acid effect of Cu, leading to the formation of a small range of acidic environments, improving the degradation efficiency of the catalyst.
H_2_O_2_ + •OH → HO_2_• + H_2_O(1)
HO_2_• + •OH → H_2_O + O_2_(2)

### 3.3. Reusability and Stability of Catalyst

From the point of view of the economic field, it is important to investigate the reusability and stability of catalysts. To examine the maintained catalytic capacity of the composite catalyst, the degradation of TC was conducted for five consecutive cycles. After each reaction, the catalyst was filtered and washed to neutral and dried, then put into the system for reuse. As shown in Figure 7, Fe_0.25_Cu_0.75_(BDC)@DE’s degradation efficiency to TC remained at 81.8% after five consecutive cycles, proving that it has good catalytic stability. 

The high TC removal over Fe_0.25_Cu_0.75_(BDC)@DE in the first round of catalytic reactions is partly related to its excellent adsorption capacity, which allows intermediates to be adsorbed on its surface during the catalytic reaction. TOC degradation efficiency within 30 min was 60%, and the removal efficiencies of TOC remained at a high level. To further demonstrate the stability of the composite catalyst, we further measured the total Fe and Cu leaching amounts in the solution. After 120 min of reaction, the concentration of total Cu satisfied the water environment discharge standards (2 mg/L) applied by the European Union, and no Fe element was found when tested in the solution.

There, the loss of catalyst activity may be due to the coverage of active sites, the accumulation of residual TC and intermediates on the catalyst surface, as well as the leakage of Cu ions from the catalyst.

### 3.4. Identification of Radicals

A spin-trapping electron paramagnetic resonance (EPR) technique and radical quenching experiments were employed to help identify free radicals generated in Fenton-like catalytic reaction systems. Quenching experiments were performed with quenchers (t-butanol (TBA) (scavenger for all •OH), p−benzoquinone (p−BQ) (scavenger for •O_2_^−^), and potassium iodide (KI) (scavenger for surface-bound radical •OH)) to determine the free radicals contributing to TC degradation. It can be seen from Figure 8a that the removal rate of TC decreased from 91.9% to 30.1% and 38.8% within 120 min after excessive addition of tert-Butanol and KI, respectively. It is proved that the degradation of TC is mainly related to the surface-bound •OH radicals, but not to the free •OH radicals, and the heterogeneous Fenton reaction played a dominant role. In addition, when excessive BQ (•O_2_^−^ radicals inhibitor) was added, the removal rate of TC decreased slightly from 91.9% to 85.5%, indicating that ·O_2_ might be involved in the degradation of TC.

For EPR tests, the spin trapping agent was DMPO (•OH), as the EPR spectrum picture shows (Figure 8b). In the ESR technique of Fe_0.25_Cu_0.75_(BDC)@DE /H_2_O_2_ system, the •OH radical has four adsorption peaks with as intensity ratio of 1:2:2:1, while there is no obvious characteristic peak in the DMPO system. The results further showed that •OH radicals were formed on the catalyst surface.

### 3.5. Reaction Pathway of TC Degradation

The degradation intermediates of TC were measured by HPLC–MS, based on this information and relevant literature [42,43]. The possible degradation pathway is presented in Figure 9. It has been observed that nine meaningful products were identified with peaks of m/z 309, m/z 324, m/z 384, m/z 249, m/z 229, m/z 283, m/z 176, m/z 131, and m/z 146 (Appendix A). Initially, the main characteristic peak of TCH (m/z = 445) was detected before degradation [44]. There are two main reasons for the generation of these intermediates throughout the Fenton-like degradation process: loss of functional groups and open-loop reactions. First, TCH (m/z = 445.0) was transformed into P1 (m/z = 309), P2 (m/z = 324), and P3 (m/z = 384). These products were mainly derived from dihydroxylation, demethylation, dehydration, and deamination products [45]. Next, the intermediates with the P4 (m/z = 249), P5 (m/z = 229) and P6 (m/z = 283) were formed due to the ring-opening reactions and the cleavage of carbon bond [46]. With further oxidization, the product with P7 (m/z = 176), P8 (m/z = 131), P9 (m/z = 146) were formed through bond cleavage, and finally some of them were completely degraded to CO_2_ and H_2_O [47].

Based on the above analysis, the dominate free radicals were confirmed and a possible mechanism of TC degradation mechanism for this heterogeneous Fenton system was proposed. Firstly, TC is adsorbed from the bulk solution to the catalyst surface via the π−π interaction between the benzene rings of TC and mesoporous DE. Then, once H_2_O_2_ is added, the metal ions on the catalyst surface combine with surface-coordinated H_2_O_2_ to initiate heterogeneous Fenton reaction. Finally, the TC molecules are attacked by •OH_surf_ radicals and mineralized into small molecular substances.

The main active substances are Fe^2+^ and Cu^+^ in activating H_2_O_2_ to produce hydroxyl •OH. Cu^+^ and Fe^2+^ are first reduced to Cu^+^ and Fe^2+^ on the catalyst surface by the captured electron from surface-coordinated H_2_O_2_ (Equations (3) and (5)). Meanwhile, Cu^+^ and Fe^2+^ lost electrons to activate H_2_O_2_ to generate •OH and Cu^2+^ and Fe^3+^ (Equations (4) and (6)). Furthermore, the standard reduction potentials of Cu (E_0_(Cu^2+^/Cu^+^) = 0.17 V) are smaller than Fe (E_0_ (Fe^3+^/Fe^2+^) = 0.77 V), Cu^+^ could promote the regeneration of Fe^2+^ through a thermodynamically profitable electron transfer process (Equation (7)). It can further promote the redox cycles of Cu^+^/Cu^2+^ and Fe^2+^/ Fe^3+^ in catalysts, and maintain the formation of •OH_surf_. Thus, the effective cyclic conversion of Fe^2+^/Fe^3+^ and Cu^+^/Cu^2+^ guarantees the continuous activation of H_2_O_2_ until it is completely consumed.
Cu^2+^+ H_2_O_2_ → Cu^+^ + HO_2_• + H^+^(3)
Cu^+^ + H_2_O_2_ +H^+^ → Cu^2+^ + •OH + H_2_O(4)
Fe^3+^ + H_2_O_2_ → Fe^2+^ + HO_2_• + H^+^(5)
Fe^2+^ + H_2_O_2_ → Fe^3+^ + •OH + OH^−^(6)
Cu^+^ + Fe^3+^ = Cu^2+^+ Fe^2+^(7)

## 4. Conclusions

We explored catalytic materials by adjusting different proportions of Fe and Cu metal elements via a facile solvothermal method. The bimetallic catalyst was then loaded on the DE to improve the stability of the catalyst, and it was used to activate H_2_O_2_ to degrade TC. Fe_0.25_Cu_0.75_(BDC)@DE/H_2_O_2_. The composite showed good catalytic performance, and the removal efficiency of TC was up to 93.0% within 120 min. Under the condition of ensuring the reusability of this catalyst, it can work efficiently in a wide pH range, and in acidic and neutral conditions. Cu has a Lewis acid effect, can form local acidic microenvironment, and can act synergically with Fe as the active site of H_2_O_2_. In addition, the degradation efficiency of the catalyst still reached 81.8% after five cycles, indicating that the catalyst has satisfactory reusability and minimal leaching of metal ions. Therefore, the stability, reusability, and high reactivity of the catalyst make it a promising application. It is possible for bimetallic MOFs to be used in the field of water remediation as pH-resistant Fenton catalysts.

## Figures and Tables

**Figure 1 nanomaterials-12-00811-f001:**
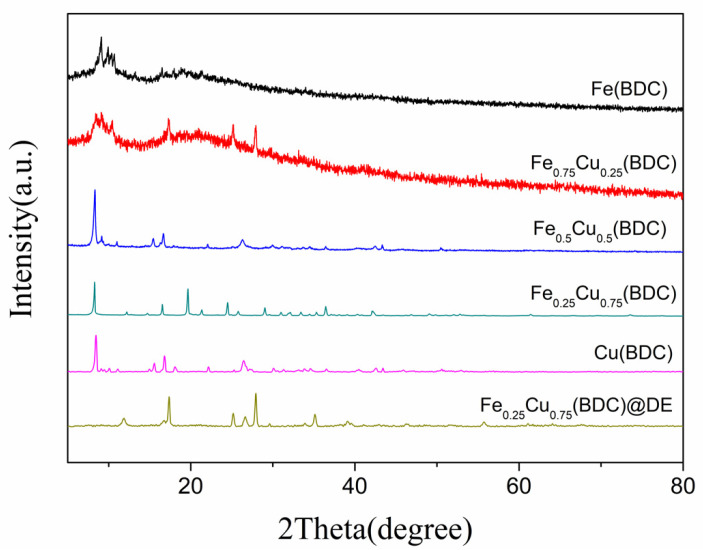
XRD patterns of Fe(BDC), Fe_0.75_Cu_0.25_(BDC), Fe_0.5_Cu_0.5_(BDC), Fe_0.25_Cu_0.75_(BDC), Cu(BDC), and Fe_0.25_Cu_0.75_(BDC)@DE.

**Figure 2 nanomaterials-12-00811-f002:**
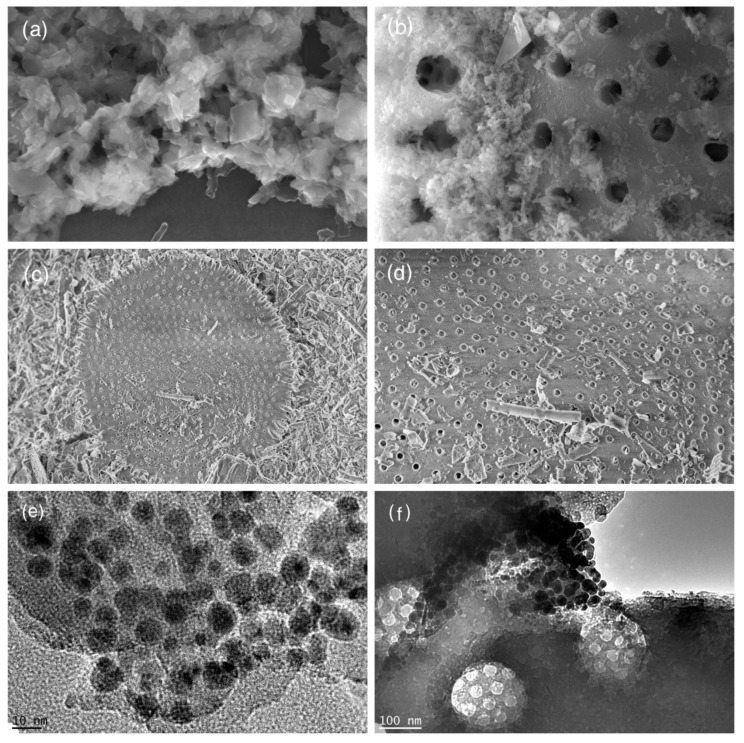
SEM images of Fe_0.25_Cu_0.75_(BDC) (**a**), Fe_0.25_Cu_0.75_(BDC)@DE (**b**) and DE (**c**,**d**); TEM images of Fe_0.25_Cu_0.75_(BDC) (**e**), Fe_0.25_Cu_0.75_(BDC)@DE (**f**).

**Figure 3 nanomaterials-12-00811-f003:**
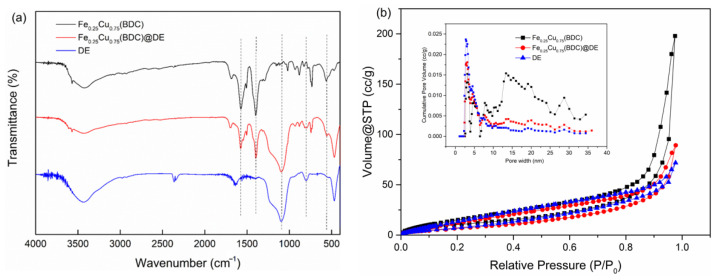
(**a**) FTIR spectra of Fe_0.25_Cu_0.75_(BDC), Fe_0.25_Cu_0.75_(BDC)@DE, DE and (**b**) N_2_ adsorption–desorption isotherms and pore size distribution plots of Fe_0.25_Cu_0.75_(BDC), Fe_0.25_Cu_0.75_(BDC)@DE, DE.

**Figure 4 nanomaterials-12-00811-f004:**
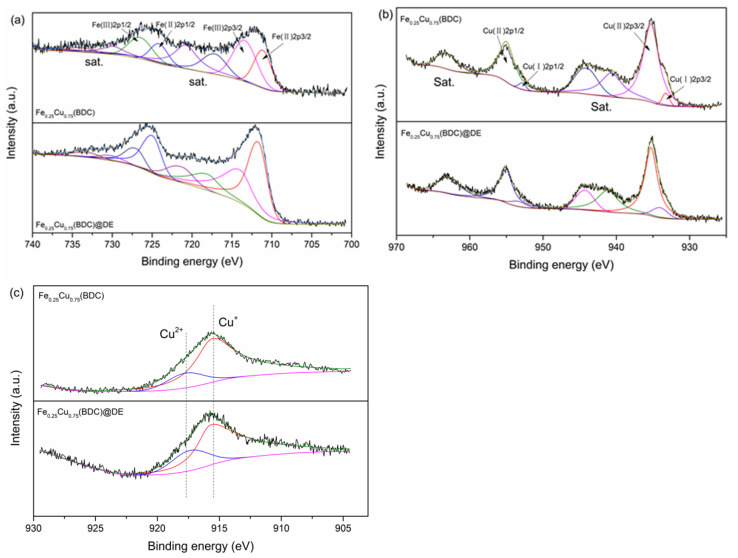
(**a**) Fe 2p XPS spectra, (**b**) Cu 2p XPS spectra, and (**c**) Cu LMM XPS spectra of Fe_0.25_Cu_0.75_(BDC) and Fe_0.25_Cu_0.75_(BDC)@DE.

**Figure 5 nanomaterials-12-00811-f005:**
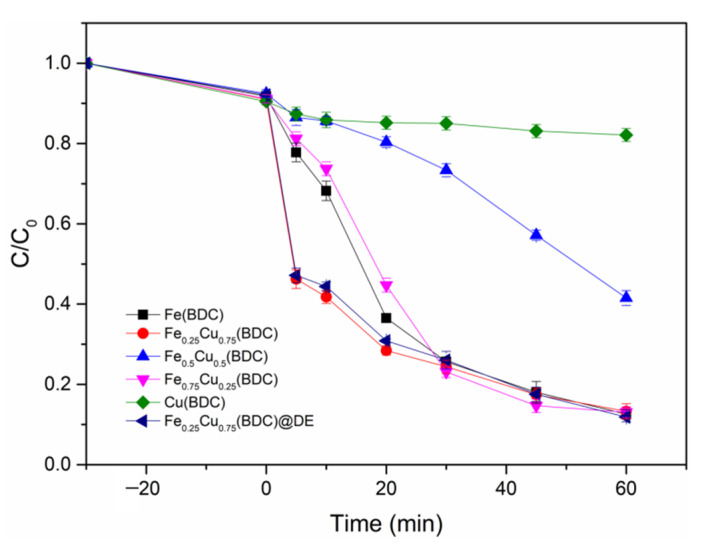
Degradation of TC by bimetallic MOFs prepared with different ratios of Fe and Cu and DE loaded MOF were compared (Experiment conditions: [TC] = 20 mg/L, [catalyst] = 0.6 g/L, [H_2_O_2_] = 10 mM, pH = 3, T = 25 °C).

**Figure 6 nanomaterials-12-00811-f006:**
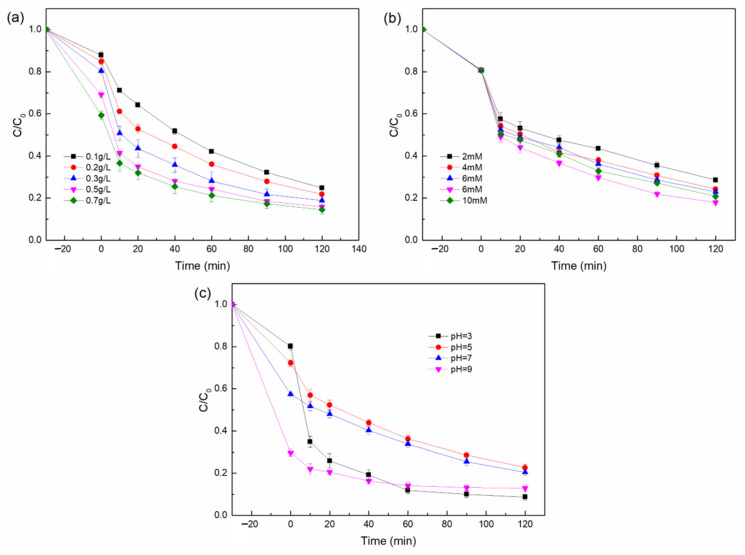
Influence factor experiments: effects of (**a**) catalyst dosage, (**b**) H_2_O_2_ concentration, and (**c**) initial pH. Except for the investigated parameter, the others were fixed: [TC] = 20 mg/L, [catalyst] = 0.5 g/L, [H_2_O_2_] = 8 mM, pH = 7, T = 25 °C.

**Figure 7 nanomaterials-12-00811-f007:**
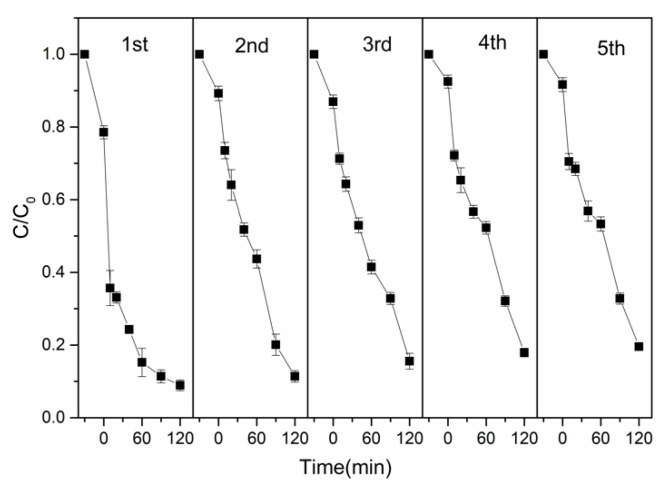
Recyclability of Fe_0.25_Cu_0.75_(BDC)@DE on the degradation of TC. (Experiment conditions: pH = 3, [TC] = 20 mg/L, [Catalyst] = 0.5 g/L, [H_2_O_2_] = 8 mM, T = 25 °C).

**Figure 8 nanomaterials-12-00811-f008:**
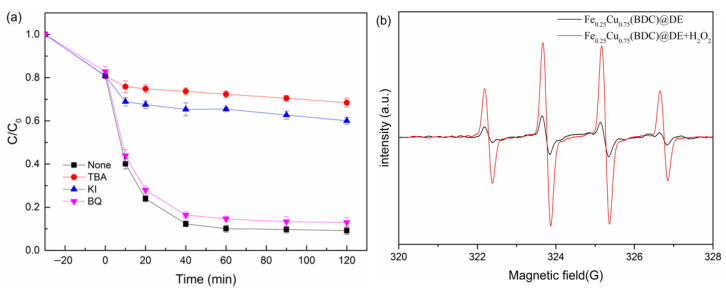
(**a**) Effect of radical scavengers on the degradation of TC. (**b**) ESR spectra of DMPO–OH· adducts that formed over time with Fenton-like degradation. (Experiment conditions: [TC] = 20 mg/L, [H_2_O_2_] = 8 mM, [catalyst] = 0.5 g/L, [BQ] = 10 mM, [TBA] = 100 mM, [KI] = 10 mM).

**Figure 9 nanomaterials-12-00811-f009:**
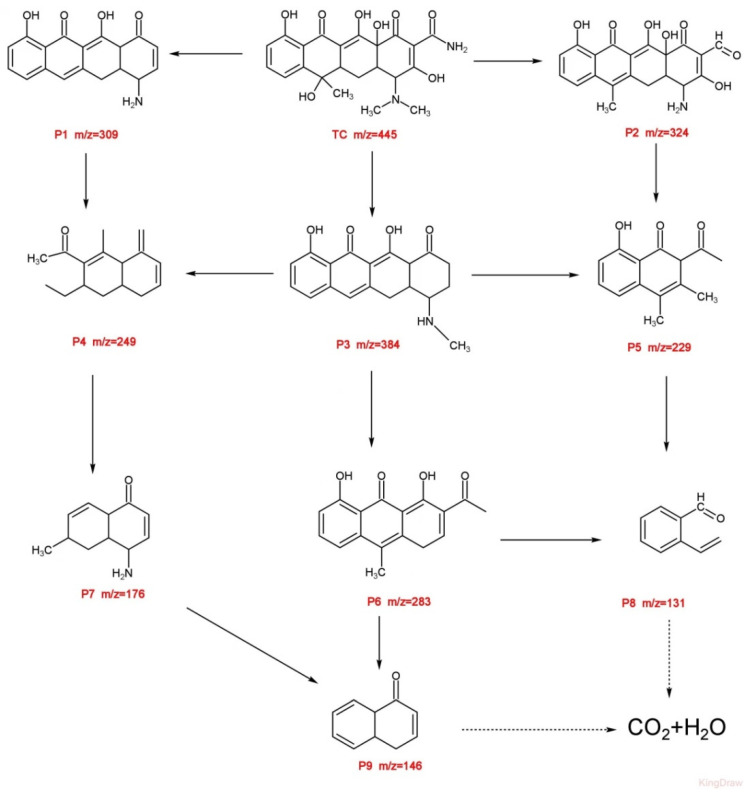
Proposed degradation pathways of TC in the Fe_0.25_Cu_0.75_(BDC)@DE /H_2_O_2_ system.

**Table 1 nanomaterials-12-00811-t001:** Synthetic parameters of all prepared samples.

Sample	N (FeCl_3_·6H_2_O) (mmol)	N (Cu(NO_3_)_2_·3H_2_O) (mmol)	N (FeCl_3_·6H_2_O)/n(Cu(NO_3_)_2_·3H_2_O)
Fe(BDC)	10.0	0	-
Fe_0.75_Cu_0.25_(BDC)	7.5	2.5	3:1
Fe_0.5_Cu_0.5_(BDC)	5	5	1:1
Fe_0.25_Cu_0.75_(BDC)	2.5	7.5	1:3
Cu(BDC)	0	10.0	-

**Table 2 nanomaterials-12-00811-t002:** Textural properties and total pore capacity of Fe_0.25_Cu_0.75_(BDC), Fe_0.25_Cu_0.75_(BDC)@DE, DE.

Sample	S_BET_ (m^2^/g)	Total Pore Volume (cm^3^/g)	Average Pore Diameter (nm)
DE	37.13	0.11	59.84
Fe_0.25_Cu_0.75_(BDC)	41.24	0.31	29.69
Fe_0.25_Cu_0.75_(BDC)@DE	30.81	0.14	17.92

## Data Availability

Not applicable.

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
