# Peer review of "Degradation of Tetracycline Hydrochloride by Cu-Doped MIL-101(Fe) Loaded Diatomite Heterogeneous Fenton Catalyst"

_nanomaterials, 2022, doi:10.3390/nano12050811_

Round 1

Reviewer 1 Report

The authors present a study on the preparation and characterization of a bimetallic catalysts loaded with diatomite, and it was used to evaluate the degradation of the tetracycline.

The investigation is well planned and elaborated by applying adequate methods. The discussion is comprehensive and conclusions are well supported by experimental results.

I recommend the manuscript to be accepted for publication after minor revision. Comments to improve the quality of the article are given below:

  • Abstract section should collect some relevant numerical results.
  • Authors need to clearly mention in the introduction part to express the novelties and objectives of this manuscript.
  • Line 11: What is the meaning of “symbol solvothermal method”?
  • The authors have to justify the concentration of the target compound used in the study.
  • For the measurement of TOC, what was the scavenger used to quench the reaction?
  • In Figure 6c and throughout the text, please change PH by pH
  • Figure 1, XRD patterns of Fe(BDC), Fe0.75Cu0.25(BDC), are of very low quality. Is it possible to improve them?

Reviewer 2 Report

Extensive synthesis & characterization of bimetallic Fe0.25Cu0.75(BDC)@DE  MOF has been reported in this article. High performance in tetracycline hydrochloride (TC) degradation achieved using this catalyst in the presence of H2O2. All the steps were nicely described. Furthermore reaction pathway of TC degradation and subsequent intermediates were characterized using HPLC-MS. Please cite following reference related to pollutant degradation in mechanism section: Chemical Engineering Journal 369 (2019) 745–757; Journal of Materials Chemistry A, 2021, 9, 5915–5951. Therefore, I recommend this article for its acceptance in Nanomaterials. 
